# AKT1 Is Required for a Complete Palbociclib-Induced Senescence Phenotype in BRAF-V600E-Driven Human Melanoma

**DOI:** 10.3390/cancers14030572

**Published:** 2022-01-23

**Authors:** Abraham L. Bayer, Jodie Pietruska, Jaymes Farrell, Siobhan McRee, Pilar Alcaide, Philip W. Hinds

**Affiliations:** 1Program in Immunology, Graduate School of Biomedical Sciences, Tufts University, Boston, MA 02111, USA; abraham.bayer@tufts.edu (A.L.B.); pilar.alcaide@tufts.edu (P.A.); 2Department of Immunology, Tufts University School of Medicine, Boston, MA 02111, USA; 3Department of Developmental, Molecular and Chemical Biology, Tufts University School of Medicine, Boston, MA 02111, USA; jodie.pietruska1@umassmed.edu (J.P.); Jaymesrfarrell@gmail.com (J.F.); skmcree@gmail.com (S.M.); 4Program in Genetics, Graduate School of Biomedical Sciences, Tufts University, Boston, MA 02111, USA

**Keywords:** melanoma, senescence, cancer, AKT, signaling

## Abstract

**Simple Summary:**

One of the hallmarks of cancer is increased cellular proliferation; therefore, many therapeutic strategies aim at arresting cellular division. Several chemotherapeutic drugs, including CDK4/6 inhibitors (CDK4/6i), result in cellular senescence, in which cells irreversibly stop proliferating and undergo many changes in function. However, some forms of senescence result in the release of factors that can worsen cancer progression. The AKT family of genes are a major regulator of cellular proliferation and senescence; therefore, we sought to study each of the three AKT isoforms in the senescence response to CDK4/6i. We found that only AKT1 is required for a full senescence phenotype induced by CDK4/6i, which works through the NF-κB pathway. This study suggests that targeting AKT1 in combination with CDK4/6i represents a therapeutic strategy in which cellular senescence can be achieved without the release of pro-cancer factors.

**Abstract:**

Cellular senescence is a carefully regulated process of proliferative arrest accompanied by functional and morphologic changes. Senescence allows damaged cells to avoid neoplastic proliferation; however, the induction of the senescence-associated secretory phenotype (SASP) can promote tumor growth. The complexity of senescence may limit the efficacy of anti-neoplastic agents, such as CDK4/6 inhibitors (Cdk4/6i), that induce a senescence-like state in tumor cells. The AKT kinase family, which contains three isoforms that play both unique and redundant roles in cancer progression, is commonly hyperactive in many cancers including melanoma and has been implicated in the regulation of senescence. To interrogate the role of AKT isoforms in Cdk4/6i-induced cellular senescence, we generated isoform-specific AKT knockout human melanoma cell lines. We found that the CDK4/6i Palbociclib induced a form of senescence in these cells that was dependent on AKT1. We then evaluated the activity of the cGAS-STING pathway, recently implicated in cellular senescence, finding that cGAS-STING function was dependent on AKT1, and pharmacologic inhibition of cGAS had little effect on senescence. However, we found SASP factors to require NF-κB function, in part dependent on a stimulatory phosphorylation of IKKα by AKT1. In summary, we provide the first evidence of a novel, isoform-specific role for AKT1 in therapy-induced senescence in human melanoma cells acting through NF-κB but independent of cGAS.

## 1. Introduction

Cellular senescence is a complex, intricate, and carefully regulated response to a variety of stressors that results in a profoundly durable arrest of cellular proliferation. First described in fibroblasts cultured for an extended time, senescence is now commonly known as an intrinsic surveillance mechanism that can delay or prevent neoplastic transformation [1,2]. This process can occur in response to DNA damage, oncogene activation, tumor suppressor gene inactivation, or in response to chemotherapy treatments. In addition to cell cycle exit, senescent cells undergo morphologic and metabolic changes that are considered mostly irreversible [3,4]. This has been attributed in part to an array of secreted proteins that reinforce the senescence phenotype, convey senescence upon neighboring cells, and drastically modify the tumor immune environment, contributing to the leukocyte-mediated elimination of senescent cells [5,6,7]. While the prevention of continuous cellular proliferation and the upregulation of pro-inflammatory signaling can serve a protective role against cancer, several features of senescent cells can in fact promote tumorigenesis. The senescence-associated secretory phenotype, or “SASP”, involves an enhanced secretion of inflammatory mediators that has been shown to drive neoplastic changes and harmful inflammation under certain circumstances [2,8,9]. This property of the SASP has led to many studies seeking pharmacologically targetable pathways to control the expression of SASP factors and for the development of “senolytics,” which enhance the elimination of senescent cells [10,11,12]. This duality of senescence influencing tumor progression underscores the importance of combinatorial therapies that could exploit the anti-proliferative aspects of senescence while moderating the pro-tumorigenic properties of the SASP.

One class of targeted therapeutics that may benefit from the modification of senescence is the CDK4/6 inhibitors (CDK4/6i). Specifically, three CDK4/6i, including Palbociclib, Ribociclib, and Abemaciclib, have been approved for the treatment of patients with hormone receptor-positive and human epidermal growth factor receptor 2-negative breast cancer and are under evaluation in clinical trials for other cancers such as melanoma [13,14,15,16]. Further, the CDK4/6i Palbociclib has been shown to induce senescence in a variety of cancer cell lines that express functional retinoblastoma protein (pRB) as well as in solid tumors in vivo [17,18,19,20,21]. Moreover, CDK4/6i have been reported to increase antigen presentation, helping to facilitate the anti-tumor immune response [22,23]. Therefore, the identification of additional common tumorigenic pathways that may reinforce cell cycle exit while supporting anti-tumor immunity could help to forestall the inevitable development of resistance to mono-targeted therapies such as Palbociclib. Melanomas are primary malignant tumors originating from melanocytes in the skin and mucosal membranes, which are often aggressive and characterized by poor prognosis [24,25]. In the case of melanoma, CDK4/6i have shown efficacy in vitro, while the success in clinical trials has been moderate or still under evaluation, making it a cancer in which the treatment could be improved by dual senolytic therapy [26,27,28].

The AKT kinase family represent compelling candidates for interaction with CDK4/6i, given their widespread functions in cellular proliferation and growth. AKT isoforms are in fact among the most frequently hyper-activated kinases in many types of cancer [29,30]. AKT activity has been shown to play a significant role in melanoma progression, with up to 20% of melanomas exhibiting loss of the AKT inhibitor PTEN [31]. Additionally, AKT activation was shown to be a strong marker of poor prognosis in patient melanoma samples [32]. The AKT family has three isoforms, AKT1, AKT2, and AKT3, which are highly homologous [30]. However, isoform selectivity has been uncovered in the targets and overall function of each isoform, especially in regard to cancer. Each AKT isoform has been found to be upregulated in different cancers, with varying impacts on tumor cell proliferation, survival, and metabolism. In melanoma, AKT1 and AKT2 activation are more commonly found in BRAF-mutant tumors, while AKT3 hyperactivity is more common in BRAF wild-type melanomas [29]. Moreover, both AKT1 and AKT2 have been implicated in melanoma metastasis [33,34]. Importantly, AKT signaling has been connected to senescence, although the isoform specificity of this observation has not been fully determined [35,36].

The convergence of CDK4/6 and AKT-regulated processes on key hallmarks of cancer strongly supports the idea that the coordinated modulation of both pathways could additively or synergistically inhibit tumor progression. Indeed, we and others have found interdependent roles for CDK4/6 and AKT in a variety of cancers [37,38,39,40]. As the spectrum of AKT activity is broad, uncovering isoform-specific AKT targets relevant in the modulation of senescence is one step towards this therapeutic approach. Importantly, recent studies have linked SASP expression to intersections between AKT, CDK4/6, and the activity of the cGAS/STING pathway—a pharmacologically relevant target that initiates innate inflammation in response to cellular infection or damage [39,41,42]. Cyclic GMP-AMP synthetase (cGAS), a cytosolic DNA sensor, has been implicated in cellular senescence and SASP induction in several studies; however, the extent to which cGAS is uniformly required in cellular senescence has yet to be determined [43,44]. The immunologic function of SASPs has further tied cGAS-STING function to the endogenous anti-tumor immune response and cancer immunotherapies, making understanding the connection between cGAS-STING signaling and other pathways regulating cancer cell survival and senescence a novel topic of interest [45,46].

Here, we show for the first time an essential, isoform-specific role for AKT1 in senescence induction in melanoma cells by Palbociclib. We also uncover an impaired cGAS-STING pathway in AKT1-deficient cells, which appears to be mostly dispensible in senescence induction in this model. Consistent with the literature, we report that AKT1 loss impairs NF-κB function, contributing to the loss of SASP induction observed here.

## 2. Materials and Methods

### 2.1. CRISPR

AKT sgRNAs were generated using sgRNA Designer (Broad Institute) or CRISPR Design (Zhang Lab, MIT) and cloned into the LentiCRISPRV2 expression plasmid. NT sgRNA sequences were taken from the GeCKO sgRNA library. Virus was produced in 293T cells, and human melanoma cell lines were infected and selected with 1 μg/mL Puromycin (Gibco, Cambridge, MA, USA) for 3 days. Pooled cells were plated at a concentration of <1 cell per well in a 96 well plate to isolate clonal cell lines. Pooled and clonal cells were validated for successful knockout by immunoblotting. See Supplemental Appendix A for sgRNA sequences.

### 2.2. Sequencing and TIDE Analysis

PCR products were generated from isolated genomic DNA of each cell line using the guide RNA of each AKT isoform. Resulting PCR products were run on a 0.8% agarose gel and extracted using a QIAquick gel extraction kit (Qiagen, Germantown, MD, USA). Sanger sequencing was submitted to Genewiz. Resulting sequence files were analyzed on the TIDE platform website (https://tide.nki.nl/, accessed on 12 March 2021) according to Brinkman et al [47].

### 2.3. Cell Culture

WM1799 or UACC903 cells were grown in DMEM media supplemented with 10% FBS (Atlanta Biologicals, Flowery Branch, GA, USA) and 1% Penicillin/Streptomycin (Invitrogen, Cambridge, MA, USA) and maintained at 37 °C. Cells were seeded in 6-well dishes at 100,000 cells/well and treated with 1 μM Palbociclib (Selleckchem, Houston, TX, USA), and/or 2.5 μM Ru521 (Selleckchem), 5 μM H151 (Selleckchem), or 0.5 μM IKK-16 (Selleckchem), for 7 days with fresh media containing treatments every 2 days. For LPS treatments, cells were seeded in 6-well dishes at 300,000 cells/well and serum starved overnight in media containing 1% FBS to minimize non-specific NF-κB signal, then treated with 1 μg/mL LPS (Invivogen, San Diego, CA, USA) for 90 min.

### 2.4. cGAS Agonist Transfection

Cells were seeded in 6-well dishes at 300,000 cells/well and serum starved overnight in media containing 1% FBS. Cells were transfected with 1 μg/mL G3-YSD (Invivogen) using Lipofectamine 3000 (Thermofisher, Waltham, MA, USA) according to the manufacturer’s instructions and harvested after 48 h.

### 2.5. CDNA Synthesis and RT-qPCR

Cells were lysed in Trizol (Invitrogen) and RNA isolated by phenol-chloroform extraction with lithium chloride precipitation. cDNA synthesis was performed using a High-Capacity cDNA Reverse Transcription Kit (ThermoFisher), and qPCR was performed using SyberGreen (ThermoFisher) on a CFX96 real-time thermal cycler (Bio-Rad, Hercules, CA, USA). See Appendix A for primer sequences.

### 2.6. Immunoblot

Cells were lysed in RIPA buffer (150 mM NaCl, 50 mM Tris pH 7.4, 1% NP-40, 0.1% SDS, 5 mM EDTA, 0.1% sodium deoxycholate, 1 mM DTT) supplemented with protease and phosphatase inhibitors (Roche), then cleared by centrifugation at 4 °C. Protein concentrations were determined using DC protein assay (BioRad), and equal concentrations of protein were subjected to SDS-PAGE then transferred to PVDF membranes, followed by blocking with Tris buffered saline + 0.1% Tween20 containing 5% non-fat milk or bovine serum albumin (BSA) for 1 h at room temperature. Immunoblotting was performed overnight at 4 °C using antibodies diluted to 1:1000 in 5% non-fat milk or BSA. Antibody sources were as folllows: AKT1 (Cell Signaling Technologies 2938), AKT2 (CST 5239), AKT3 (CST 8018), pAKT1 (CST 9018), pAKT2 (CST 8599), pan pAKT (CST 4060), pan AKT (CST 4691), GAPDH (CST 2118), RB (BD Biosciences G3425), pRb (CST 8180), α-tubulin (CST 3873), β-actin (Sigma A5441) pP65 (CST 3033), P65 (CST 4764), p105/p50 (CST 3035), cGAS (CST 15102), STING (CST 50494), pIRF3 (CST 4947), IRF3 (CST 4302), IKKα (CST 2682), pIKKα/β (CST 2697), pIKKα T23 (abcam 38515) ERK (CST 9102). Blots were then incubated with the appropriate HRP-peroxidase conjugated secondary antibody (1:2000, Rabbit CST 7074, Mouse CST 7076) at room temperature for 1 h, then developed using Pierce ECL (ThermoFisher). Band intensity was quantified using Image-J.

### 2.7. Beta-Galactosidase Staining

Cells were fixed in PBS at pH 6.0 containing 0.5% glutaraldehyde for 5 min, then washed with PBS. Cells were then incubated in phosphate buffer at pH 6.0 containing 5 mM potassium ferrocyanide, 5 mM potassium ferricyanide, 1 mM MgCl_2_, and 1 mg/mL X-Gal (Thermofisher) for 24 h at 37 °C. Cells were washed in PBS, then mounted in 90% glycerol, and then imaged on an Eclipse 80i microscope using SPOT image acquisition software.

### 2.8. 2′3′-cGAMP ELISA

Cells were transfected with G3-YSD or treated with Palbociclib as previously described. Pellets were lysed in RIPA buffer supplemented with protease inhibitors (Roche, Basel, Switzerland), then cleared by centrifugation at 4 °C. Equal concentrations of protein were used for each sample, and the ELISA was run as described by the manufacturer (Cayman Chemical, Ann Arbor, MI, USA).

### 2.9. Cell Cycle Analysis

Cells were cultured and treated with Palbociclib for 24 or 48 h as previously described. Cells were collected and washed with cold PBS, then fixed in ice-cold 70% EtOH for 30 min at 4 °C. Cell pellets were washed in staining buffer (PBS no Ca/Mg, 3% FBS, + mM EDTA), then incubated in staining buffer with 80 μg/mL Propidium Iodide and 0.125 mg/mL RNAse A (Thermofisher R1253) for 40 min at 30 °C. Samples were run on a BD LSRII (BD Biosciences, Franklin Lakes, NJ, USA) and analyzed using FlowJo software.

### 2.10. Proliferation Assay

Cells were cultured and treated with DMSO or Palbociclib as previously described. After 7 days of Palbociclib or DMSO treatment, cells were incubated in 10 μM carboxyfluorescein diacetate succinimidyl ester (CFSE, Thermofisher V128830) in PBS for 15 min at 37 °C. Dye was removed and replaced with fresh media containing Palbociclib or DMSO for 72 h. Cells were collected and fixed for 20 min at RT with fixation buffer (Biolegend 420801, San Diego, CA, USA), then samples were run on a BD LSRII (BD Biosciences) and analyzed using FlowJo software.

### 2.11. Statistical Analysis

Data are expressed as mean ± SEM unless otherwise indicated. Statistical analyses between two groups were performed by the unpaired Student’s *t*-test. Multiple group comparisons were performed by one or two-way ANOVA with the Sidák post-test where indicated. The difference was considered statistically significant at * = *p* < 0.05, ** = *p* < 0.01, and *** = *p* < 0.001. All statistical analyses were performed using GraphPad Prism.

## 3. Results

### 3.1. AKT1 Is Required for Aspects of Palbociclib-Induced Senescence in WM1799 Cells

CDK4/6 and the AKT family of enzymes collaborate in a number of key cancer hallmarks, leading to the possibility that combined inhibition could be more efficacious as an anti-neoplastic therapy than either alone. However, pan-AKT inhibition is problematic in vivo due to numerous impacts on normal tissue and function. Therefore, the identification of AKT isoform-specific functions may support the targeting of differentially regulated pathways by individual AKT isoforms in combinatorial therapeutic approaches. In order to study isoform-specific, anti-neoplastic effects of AKT in human melanoma cells, we sought to knockout each AKT isoform individually. Using isoform-specific guide RNAs (gRNAs), we used CRISPR-Cas9 to knockout AKT1, AKT2, or AKT3 in multiple melanoma cell lines, validating the success of AKT knockout by Western blotting [48]. WM1799 cells, a human BRAF V600E mutated melanoma cell line, appeared to tolerate AKT knockout well and showed complete knockout of each isoform based on immunoblotting (Figure 1A).

To test if AKT isoforms played different roles in CDK4/6 inhibition-induced cellular senescence, we treated each cell line with Palbociclib (PALBO) for 7 days and evaluated senescence by determining beta-galactosidase activity using cell staining. All cells treated with PALBO showed morphologic changes and a significantly reduced cell number. Interestingly, we observed that only AKT1 knockout cells exhibited an impaired senescence phenotype as seen by a lack of significant pigment accumulation in most cells upon X-gal staining compared to a non-targeting guide (NT) or AKT2/AKT3 knockout cells (Figure 1B,C). We then tested if AKT1 played a similar role in SASP induction. Senescent cells were evaluated for the expression of candidate SASP factors IL6 and IL8 by RT-qPCR, which were strongly induced by PALBO treatment in each cell line except AKT1 knockout cells, indicating that full SASP elaboration is also dependent on AKT1 (Figure 1D,E). UACC903 cells, a different human BRAF mutant melanoma cell line, showed a similar lack of beta-galactosidase staining and SASP induction in AKT1 cells, with a modest effect in AKT3 knockout cells as well, showing this phenotype is not cell-line specific (Figure 1F–I). Together, these data show an isoform-specific AKT1 requirement in key aspects of Palbociclib-induced senescence, including the induction of several SASP factors. Despite this phenotype, AKT1 appears to be dispensable for the Palbociclib-induced arrest of proliferation.

### 3.2. Clonal AKT1 Knockout Cells Exhibit Impaired Senescence and SASP Induction

Due to the importance of AKT1 in cellular proliferation, viable clonal cell lines were very slow to grow. Therefore, our initial senescence phenotype was established using pooled CRISPR-treated cell lines, which would also mitigate aspects of potential clonal heterogeneity in these human tumor cell lines. We analyzed these cell lines using DNA sequencing and TIDE analysis, which showed >90% AKT1 mutation resulting from the presence of multiple distinct mutation events (Appendix A). Because the CRISPR-induced genetic heterogeneity of the pooled cell line raised the possibility of phenotypic heterogeneity, after our initial screen showed an AKT1-specific effect on senescence, we generated single-cell dilution clones from the NT and AKT1 knockout WM1799 cell pools. We obtained two AKT1 clones exhibiting a frameshift mutation that completely abolished AKT1 expression by Western blot as well as three NT clones with no effect on AKT1 (Appendix A). Several other clones isolated that did not show AKT1 loss at the protein level were excluded from studies moving forward.

Considering the striking phenotype in pooled cell lines, we anticipated that clonal cell lines would reproduce similarly impaired senescence and SASP. We treated NT and AKT1 WM1799 clones with PALBO for 7 days, then analyzed senescence as previously performed. Similarly to the pooled cell lines, the AKT1 knockout clonal cell lines exhibited a striking lack of beta-gal staining (Figure 2A,B). To further support our observed phenotype, we evaluated beta-gal staining following PALBO treatment in all clonal cell lines. Three isolated NT clones showed strong beta-gal staining, while both complete AKT1 knockout clones showed no signal after 7 days of treatment (Appendix A–C). Similar to the pooled cell lines, we found a strong induction of ΙL6 and IL8, as well as IL-1β after PALBO treatment, which was completely abrogated with AKT1 knockout (Figure 2C–E). To test if SASP induction correlated with either AKT1 activity or that of another isoform, we probed for Serine 473/4 phosphorylation in NT cells treated with PALBO—a simple marker of kinase activity. To our surprise, we found that PALBO-treated cells exhibited lower AKT1, AKT2, and total phosphorylation at these residues, suggesting a much more complex regulatory mechanism of AKT activity by CDK4/6 inhibition (Figure 2F). However, together, these data solidify a significant role of AKT1 in the induction of senescence and SASPs.

To confirm that the phenotype observed in AKT1 knockout cells was specific to senescence and not a difference in proliferative arrest, we performed a cell cycle analysis by evaluating propidium iodide incorporation. We found that NT and AKT1 knockout cells showed a similar arrest, as seen by the percentage of cells in G1 phase, after 24 or 48 h of treatment with PALBO (Figure 2G,H). We further evaluated Retinoblastoma protein (pRb) phosphorylation, the target of CDK4/6, and show that in both NT and AKT1 knockout cells, Rb phosphorylation and total protein was equally decreased, indicating the effectiveness of PALBO in each cell line (Figure 2I). These data confirm that PALBO results in cell cycle exit independent of AKT1. We then asked if the proliferative arrest was irreversible following the removal of PALBO, as would be expected in senescent cells, and if AKT1 loss impacted this. Using CFSE tracing to monitor proliferation, we found that in both NT and AKT1 knockout cells, 3 days after the removal of PALBO, only around 5% of cells had begun proliferating, compared to over 60% of untreated cells, indicating a mostly irreversible cell cycle arrest (Figure 2J,K). These data support our conclusion that AKT1 is required specifically for the induction of key aspects of the senescence phenotype by PALBO but is dispensable for irreversible proliferative arrest.

### 3.3. cGAS-STING Is Impaired in AKT1 Knockout Cells

Given the newfound importance of the DNA sensor cGAS in senescence [43,49], as well as the discovery that cGAS is an AKT substrate [41], we evaluated this pathway as a mechanism for impaired senescence. As cGAS-STING is often inactivated in cancer, we first tested if the pathway was functional in WM1799 cells. We confirmed the presence of both cGAS and STING at the transcript and protein level, interestingly finding a significantly lower level of cGAS but higher level of STING in AKT1 knockout cells (Figure 3A, Appendix A). To evaluate the functional capability of cGAS-STING, we transfected both cell lines with G3-YSD, a short, double-stranded hairpin DNA sequence known to activate cGAS, then assessed the functionality of the pathway at multiple points. In NT cells, transfection resulted in an increase in IRF3 S396 phosphorylation, the main downstream target of STING activation, with no increase observed in AKT1 knockout cells (Figure 3A). Additionally, we found that the type I interferons IFNα and IFNβ, a direct target of IRF3, were both induced only in NT cells at the transcript level after dsDNA transfection (Figure 3B,C). We also measured 2′3′cGAMP by ELISA, a direct functional readout of cGAS activity, and only detected cGAMP in the cytosol of transfected NT cells (Figure 3D). This suggests that WM1799 cells have a functional cGAS-STING pathway, which is dependent on the presence of AKT1. Due to the striking loss of cGAS protein and decrease in mRNA level in AKT1 knockout cells, we hypothesize that AKT1 promotes cGAS expression, leading to decreased cGAS function in AKT1 knockout cells, with a potential compensatory increase in STING levels.

### 3.4. cGAS/STING Inhibition Does Not Affect Senescence or SASP Secretion

To begin to evaluate the importance of the cGAS-STING pathway in senescence, we first examined the effect of PALBO on cGAS and STING expression and activity. We found that at the transcript and protein levels, PALBO treatment significantly decreased cGAS independent of AKT1, whereas PALBO did not appear to affect levels of STING (Appendix A–C). Additionally, no cGAMP was detectable in PALBO-treated NT or AKT1 knockout cells, suggesting that Palbociclib is not significantly inducing the activity of cGAS.

To further evaluate if cGAS-STING plays a role in Palbociclib-induced senescence, we first measured the expression of type I interferons following PALBO treatment. Surprisingly, we found a strong induction of IFNα and IFNβ by PALBO in WM1799 cells, which became significant with AKT1 loss, suggesting these SASP factors may expressed through cGAS independent STING activity (Figure 3E,F). To separate the contributions of cGAS vs. STING in Palbociclib-induced senescence, we inhibited cGAS and STING separately using Ru521, a small molecule inhibitor of cGAS, or H151, a small molecule inhibitor of STING. First, we confirmed that these inhibitors were functional by pre-treating NT cells with each, then transfecting with G3-YSD as previously performed. Each inhibitor blunted the induction of IFNα and IFNβ (Appendix A), confirming that these agents impair cGAS-STING signaling in these cells. We then treated cells with each inhibitor in combination with PALBO for 1 week, then analyzed senescence as described previously. Neither the inhibition of cGAS or STING altered beta-galactosidase staining in PALBO-treated samples (Figure 3G,H). Interestingly, the inhibition of cGAS had little effect on SASP expression in NT cells (Figure 3I–K). The inhibition of STING, however, prevented the statistically significant increase in interleukin SASP factors seen with PALBO treatment alone in NT cells (Figure 3I–K) and completely eliminated IFN induction in both NT and AKT1 knockout cells (Figure 3E,F). Together, this suggests that cGAS does not contribute significantly towards the senescence phenotype, whereas STING contributes towards SASP factor expression independent of cGAS.

### 3.5. NF-κB Function Regulates SASP Factors in an AKT1-Dependent Manner

Given our observation that Palbociclib-induced expression of interleukins, but not interferons, was AKT1 dependent and cGAS independent, we next focused on NF-κB function as a potential contributor to Palbociclib-induced SASP factor expression, as has been observed in other settings [50,51]. First, we determined if PALBO induced NF-κB in WM1799 cells. We observed an AKT1-dependent increase in p65 phosphorylation in PALBO-treated cells, indicative of a role for this pathway in SASP induction and therefore senescence (Figure 4A). To test if NF-κB function was generally impaired with AKT1 knockout, we treated each cell line with lipopolysaccharide (LPS), a Toll-like receptor agonist which activates NF-κB. Similarly to PALBO treatment, we found an AKT1-dependent increase in P65 phosphorylation with LPS treatment, as well as lower levels of the p100/p50 subunits of NF-κB in AKT1 knockout cells (Appendix A–C). This confirmed that AKT1 knockout cells have defective NF-κB signaling.

AKT is known to promote NF-κB function by phosphorylating IKK [52]; therefore, we hypothesized that the loss of this activating factor dampened IKK activity and impaired SASP induction. Threonine 23 on IKKα contains an AKT consensus site and is a known substrate of AKT (Figure 4B); therefore, we checked phosphorylation at this site by immunoblot. As expected, we found that PALBO induced phosphorylation at T23 on IKKα in NT cells only (Figure 4B). Interestingly, Ser176 on IKKα/β showed a similar increase in phosphorylation in both cell lines in response to PALBO. This suggests that the phosphorylation of T23 by AKT1 is a “fine tuning” mechanism for IKK function independent of Ser176 phosphorylation, and that both are required for SASP elaboration in Palbociclib-induced senescence.

To further confirm the importance of IKK and the NF-κB pathway in Palbociclib-induced senescence, we used an IKK inhibitor (IKK-16) in combination with PALBO treatment. Co-treatment of NT cells with PALBO and IKK-16 for 1 week resulted in reduced beta-galactosidase staining (Figure 4C,D). Additionally, IKK-16 treatment completely eliminated both IL-1β and IL8 induction by PALBO (Figure 4E–G). Interestingly, IL6 induction was not affected, suggesting that multiple Palbociclib-responsive mechanisms may induce this key component of the SASP. However, we saw a suppression of P65 phosphorylation by IKK-16 in both NT and AKT1 cells treated with PALBO—evidence that the inhibitor was working as expected (Figure 4H). Finally, to determine if the induction of NF-κB by PALBO was solely dependent on AKT1 or required another isoform, we treated AKT2 and AKT3 knockout WM1799 cell lines with PALBO and found that both retained the induction of P65 phosphorylation after treatment (Figure 4I). Together, these data together confirm that Palbociclib-induced senescence is in part mediated through IKK function dependent on the presence and activity of AKT1. A scheme of the mechanisms relating NF-κB, cGAS, STING, and AKT1 supported by our data is shown in Figure 5.

## 4. Discussion

Here, we show for the first time an AKT1 isoform-specific role in cellular senescence induced by CDK4/6 inhibition in human melanoma cell lines. The striking lack of beta-galactosidase staining and absence of multiple SASP factors implies a requirement for AKT1 in the induction of these aspects of senescence by Palbociclib; in contrast, AKT1 loss does not impact the irreversible growth arrest typical of senescent cells that is induced by Palbociclib treatment. The AKT family has been connected to senescence in the past, including one study showing the MTOR-AKT axis to be impacted by PALBO treatment. Interestingly, similar to our study, decreased AKT phosphorylation was observed, but no effect of AKT over-expression on the senescence phenotype was seen, perhaps due to compensatory changes in activity of individual isoforms [35,53]. This previous work considers all AKT isoform activity together, highlighting the novelty of our study, in which we compare all three AKT isoforms independently to fully determine the isoform specificity in senescence. It is important to note that our study is limited to a form of therapy-induced senescence instigated by CDK4/6 inhibition, and therefore a role for AKT1 in other forms of senescence, such as replicative senescence, oncogene-induced senescence, or therapy-induced senescence by other agents, cannot be assumed without further studies of these systems. Our study is also limited by not performing the overexpression rescue experiment; however, based on the lack of phenotype observed in previous studies and the difficulty in the interpretation of AKT overexpression in in vitro models, we chose to focus on characterizing senescence in the setting of genetic deletion.

As a potential mechanism explaining the dependence of Palbociclib-induced senescence on AKT1, we explored whether AKT1 impacted cGAS-STING function—a newly discovered player in senescence [43,49]. We show here that AKT1 loss reduces cGAS at the protein level and results in impaired pathway functionality in response to dsDNA transfection. This finding is surprising, considering an elegant study by Seo et al. in which they outline a suppressive AKT phosphorylation site on cGAS [41]. While the loss of AKT1 might be expected to enhance cGAS function by the reduction in the phosphorylation that Seo et al. describe, we found the opposite, showing 2′3′-cGAMP production in response to cGAS agonism to be dependent on the presence of AKT1. However, while we did test the functionality of cGAS and STING at multiple points, we did not examine cGAS phosphorylation sites directly. This leaves open the possibility that AKT2 or AKT3, both of which are expressed in AKT1 knockout WM1799 cells, may be activated to suppress cGAS function after the loss of AKT1.

Using the pharmacologic inhibition of cGAS, we found no changes in senescence as measured by SA beta-gal staining, nor did we observe any impairment of SASP factor production. Given recent publications implicating cGAS in senescence, including senescence induced by CDK4/6 inhibition, this finding was unexpected [43,49]. It is possible that the pharmacologic inhibition of cGAS is inefficient compared to genetic disruption and that the residual activity of cGAS can increase STING activity sufficiently for SASP factor expression, but not for dsDNA induction of IFNs. However, it is equally plausible that a cGAS-independent mechanism of STING activation functions in response to CDK4/6 inhibition. For example, dsDNA, recognized by other cytosolic DNA sensors, could lead to the activation of STING and IKK through the TRAF-STING-TBK1 axis, as shown by Abe and Barber in immortalized fibroblasts [54]. Future studies focused on how Palbociclib directly activates this response and which substrates are modulated by AKT1 in addition to IKKα may provide opportunities for the therapeutic modulation of the SASP in situations where therapy-induced senescence impacts anti-neoplastic responses.

In this regard, we confirm here a role for AKT1-dependent NF-κB signaling in SASP induction by using the pharmacologic inhibition of IKK. Based on previous work showing that AKT could enhance IKK function by phosphorylating IKKα, we showed an increase in phosphorylation at a recognized AKT phosphorylation site on IKKα induced by Palbociclib in WM1799 NT cells but not AKT1 knockout cells. Moreover, the pharmacologic inhibition of IKK impairs Palbociclib-induced senescence in these cells and attenuates SASP factor expression, with the exception of IL6. However, IL6 expression remains dependent on AKT1 and STING, suggesting that the regulation of individual SASP factors in response to CDK4/6 inhibition in melanoma cells is complex, with compensatory mechanisms we have yet to elucidate. Therefore, while the roles of AKT1 and IKK in modulating the SASP response to Palbociclib represent important components of this form of therapy-induced senescence, the full details of the connection between CDK4/6 inhibition, AKT1 function, and senescence remain unknown (Figure 5). Additionally, as we found decreased S473/S474 phosphorylation of AKT isoforms induced by PALBO, we hypothesize that PALBO might induce non-canonical kinase activity that results in IKK phosphorylation and subsequent SASP induction; however, this will be addressed in a future study. Future work could also include a multi-omics study to give a clearer picture of the changes induced by Palbociclib treatment in melanoma cells compared to other tumor cell types that also undergo senescence-like changes in response to this important anti-neoplastic agent. In any case, the data that we present here suggest that Palbociclib in combination with inhibitors of factors including AKT1, STING, or IKK may provide an eventual strategy to modulate the SASP-related, inflammatory environment that could augment anti-tumor responses in a context-dependent manner.

## 5. Conclusions

In conclusion, we have demonstrated an essential and isoform-specific role for AKT1 in the induction of senescence by CDK4/6 inhibition, as well as in cGAS-STING function. We attribute this to impaired NF-κB signaling in SASP induction. Our work shows yet another example of isoform specificity in AKT function and provides support for AKT1 isoform-specific inhibition as a novel therapeutic target.

## Figures and Tables

**Figure 1 cancers-14-00572-f001:**
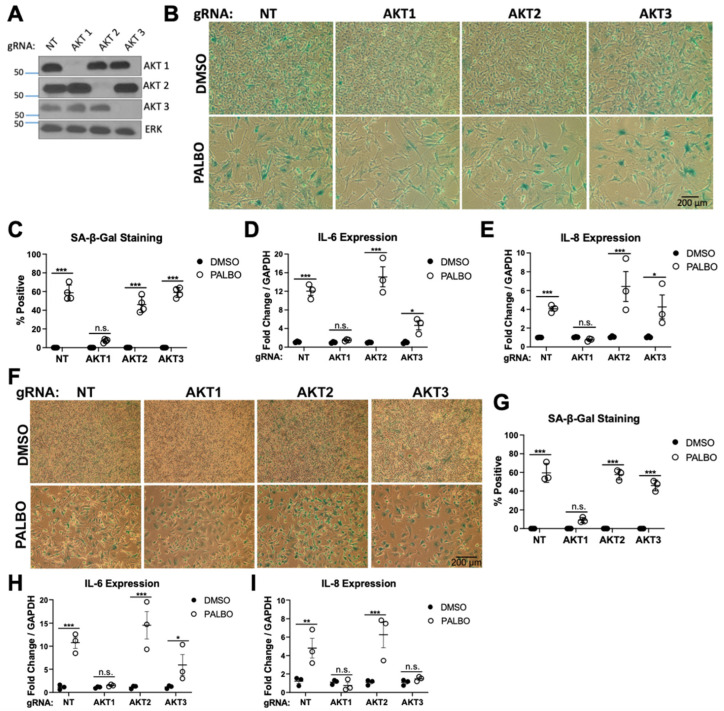
Only AKT1 is required for Palbociclib-induced senescence in melanoma cell lines (**A**). Western blot showing knockout of each AKT isoform specific to each guide RNA (gRNA) compared to non-targeting gRNA (NT) and untreated cells (**B**). Beta-galactosidase staining of DMSO or PALBO treated cells, quantified in (**C**). (**D**,**E**). Induction of IL6 and IL8 by RT-qPCR in pooled cells after 7 days of treatment with Palbociclib (PALBO) at 1 μM or DMSO as a control. (**F**). Beta-galactosidase staining in UACC903 cells of after 7 days of treatment with PALBO at 1 μM or DMSO as a control, quantified in (**G**). (**H**,**I**). Induction of IL6 and IL8 by RT-qPCR in PALBO or DMSO-treated cells. Error bars represent SEM. Significance determined by Student’s *t*-test (* = *p* < 0.05, ** = *p* < 0.01, *** = *p* < 0.001, n.s. = no significance). The uncropped blots are shown in Appendix A.

**Figure 2 cancers-14-00572-f002:**
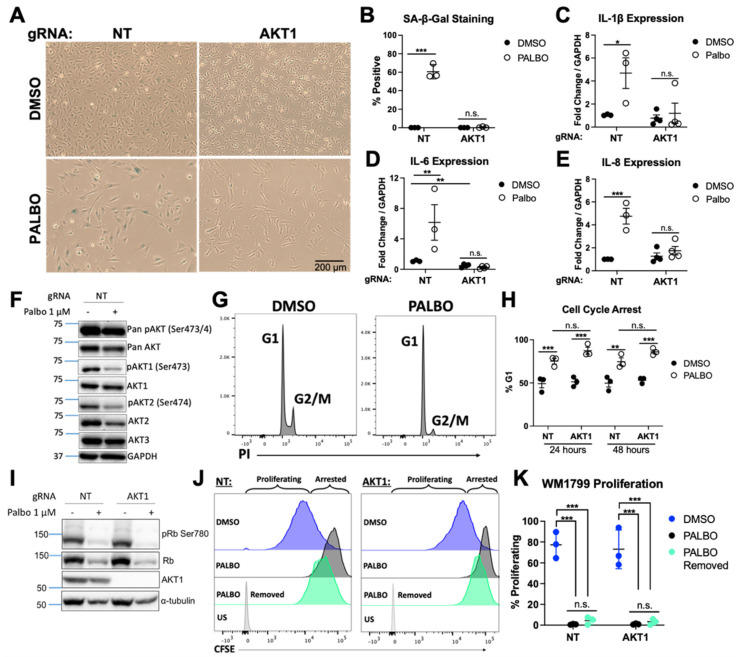
AKT1 knockout clones exhibit impaired senescence and SASP production (**A**). Beta-galactosidase staining of 1 μM Palbociclib (PALBO) or DMSO-treated clonal cell lines for 7 days, quantified in (**B**). (**C**–**E**). Induction of IL1, IL6, and IL8 in clonal cell lines by RT-qPCR after 7 days of treatment with PALBO at 1 μM or DMSO as a control (**F**). Representative Western blot of cells treated with PALBO or DMSO for 7 days for individual and total AKT phosphorylation. (**G**) Representative flow cytometry histograms of PI-stained cells for cell cycle analysis treated with PALBO or DMSO for 24 h. (**H**) Percentage of cells arrested in G1 phase after 24 or 48 h of PALBO treatment, quantified by flow cytometry after PI staining. (**I**) Representative Western blot of cells treated with PALBO after 7 days for retinoblastoma proteins phosphorylated on residue Ser780 (pRb) vs. total retinoblastoma protein (Rb). (**J**). Representative flow cytometry plots of CFSE-stained cells to evaluate proliferation, after treatment with DMSO and PALBO for 7 days, or PALBO for 7 days with subsequent drug removal for 72 h prior to analysis, quantified in (**K**). Error bars represent SEM. Significance determined by Student’s *t*-test, or two-way ANOVA with multiple comparisons (* = *p* < 0.05, ** = *p* < 0.01, *** = *p* < 0.001, n.s. = no significance). The uncropped blots are shown in Appendix A.

**Figure 3 cancers-14-00572-f003:**
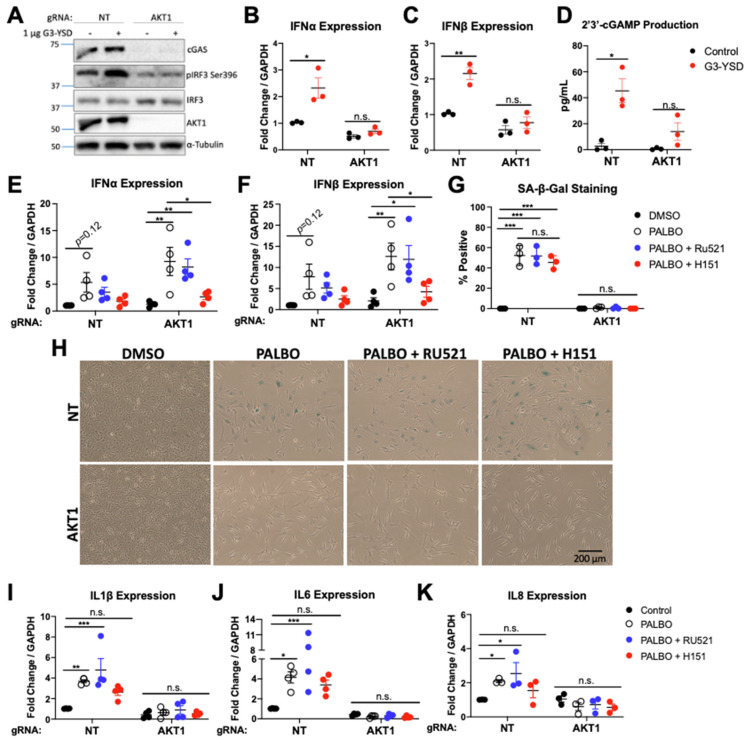
AKT1 is required for cGAS/STING function independent of Palbociclib-induced senescence. (**A**). Western blot of cGAS and IRF3 phosphorylation 48 h after transfection with 1 μg/mL dsDNA (G3-YSD). (**B**,**C**). Induction of IFNα/β by RT-qPCR 48 h after transfection with G3-YSD. (**D**). ELISA for 2′3′-cGAMP from lysates of transfected cells. (**E**,**F**). Induction of IFNα/β by RT-qPCR after 7 days of treatment with Palbociclib (PALBO) at 1 μΜ, PALBO with cGAS inhibitor (Ru521) at 2.5 μΜ, or PALBO with STING inhibitor (H151) at 5 μΜ (**G**,**H**). Beta-galactosidase staining of cells treated for 7 days with PALBO (1 μΜ) alone or in combination with cGAS inhibitor Ru521 (2.5 μΜ) or STING inhibitor H151 (5 μΜ). (**I**–**K**). SASP induction after treatment with PALBO and cGAS/STING inhibitors. Error bars represent SEM. Significance determined by Student’s *t*-test (Panel (**B**–**D**)) or two-way ANOVA with multiple comparisons (Panel (**E**–**K**)) (* = *p* < 0.05, ** = *p* < 0.01, *** = *p* < 0.001, n.s. = no significance). The uncropped blots are shown in Appendix A.

**Figure 4 cancers-14-00572-f004:**
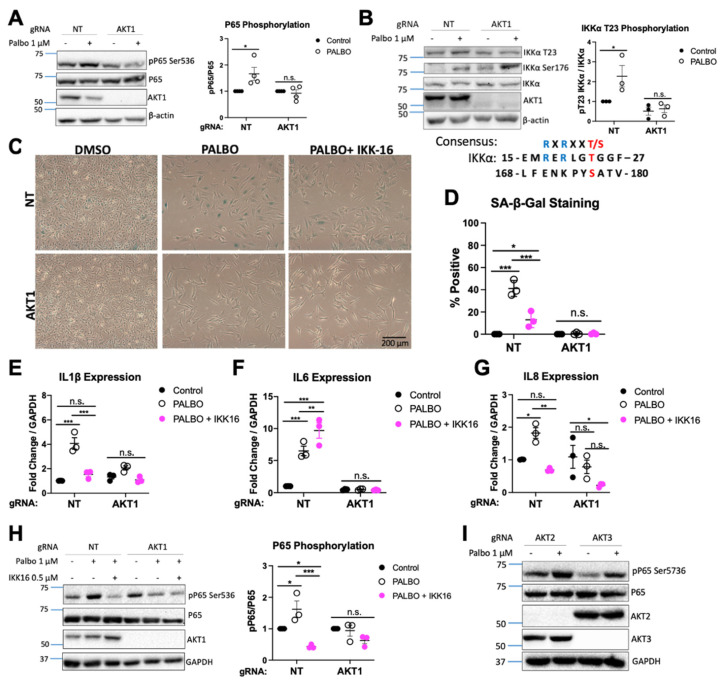
AKT1 regulates IKK leading to NF-κB-mediated SASP production (**A**). Representative Western blot of cells treated with PALBO for 7 days at 1 μM with quantification of phospho-P65 over total P65 normalized to untreated cells for *n* = 4 independent experiments. (**B**). Representative Western blot of IKKα phosphorylation of cells treated with PALBO for 1 week at 1 μM with quantification of phospho-P65 over total P65 normalized to untreated cells for *n* = 3 independent experiments, AKT consensus site at T23 aligned with S176 in IKKα. (**C**). Beta-galactosidase staining of cells treated for 7 days with PALBO (1 μΜ) alone or in combination with IKK inhibitor IKK-16 (0.5 μΜ). (**D**). Representative Western blot of NT or AKT1 cells treated for 7 days with PALBO alone or in combination with IKK-16. (**E**–**G**). SASP induction in cells treated with PALBO alone or in combination with IKK-16 for 1 week. (**H**). Representative Western blot of cells treated with PALBO (1 μΜ) alone or in combination with IKK inhibitor IKK-16 (0.5 μΜ) with quantification of phospho-P65 over total P65 normalized to untreated cells. (**I**). Representative Western blot of bulk AKT2/3 KO cells treated with PALBO for 7 days at 1 μM. Significance determined by Student’s *t*-test (Panel (**A**,**B**)) or two-way ANOVA with multiple comparisons (Panel (**E**–**G**)) (* = *p* < 0.05, ** = *p* < 0.01, *** = *p* < 0.001, n.s. = no significance). The uncropped blots are shown in Appendix A.

**Figure 5 cancers-14-00572-f005:**
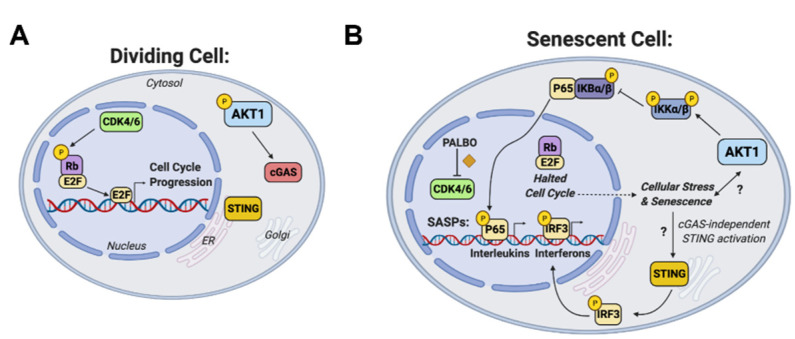
Scheme of role of AKT1 in regulation of senescence, cGAS-STING, and IKK function. (**A**) Dividing cell in which CDK4/6 phosphorylation of Rb contributes to E2F-mediated cell cycle progression, and AKT1 promotes cGAS expression. (**B**) Senescent cell in which Palbociclib (PALBO) inhibits CDK4/6, halting cellular proliferation and contributing to cellular stress and the senescence response through AKT1-mediated NFkB action and cGAS-independent STING activity. Previously undefined abbreviations: CDK = cyclin dependent kinase, Rb = retinoblastoma protein, E2F = E2 transcription factor, ER = endoplasmic reticulum.

## Data Availability

Data are available within the article or on request from the authors.

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
