# Peer review of "AKT1 Is Required for a Complete Palbociclib-Induced Senescence Phenotype in BRAF-V600E-Driven Human Melanoma"

_cancers, 2022, doi:10.3390/cancers14030572_

Round 1
Reviewer 1 Report
It is a good, well-written manuscript. I only have small remarks:
- Line 135: Why the cells were serum-starved for LPS treatment?
- Proliferation assay: What was the final concentration of DMSO during the treatment?
- For the Celsius degrees, the Authors should use °C symbol, not only C (e.g. lines 153, 158, 173, 185, 187).
- Line 179: What kind of degrees? Celsius?
Author Response
It is a good, well-written manuscript. I only have small remarks:
Response: Thank you for your kind feedback.
1. Line 135: Why the cells were serum-starved for LPS treatment?
Response: The cells were serum-starved with 1% FBS to minimize non-specific NF-kB signal that is induced by the combination of proteins, growth factors, and cytokines that are in FBS, as commonly done. This helps to see the NF-kB induction specifically by LPS (or lack thereof in AKT1 deficient cells). This is mentioned in the methods briefly now (lines 145-156).
2. Proliferation assay: What was the final concentration of DMSO during the treatment?
Response: The final concentration of DMSO is 0.02%, an equivalent volume of DMSO as added to the PALBO treated conditions, as PALBO is dissolved in DMSO.
3. For the Celsius degrees, the Authors should use °C symbol, not only C (e.g. lines 153, 158, 173, 185, 187).
Response: The text has been updated accordingly.
4. Line 179: What kind of degrees? Celsius?
Response: The text has been updated accordingly to say 4°C
Reviewer 2 Report
A very interesting original paper demonstrating an essential and isoform-specific role for AKT1 in the induction of senescence by Cyclin-dependent kinase 4/6 inhibition, and in the cGAS-STING function, suggesting the inhibition of AKT1 as a novel therapeutic target. Only minor revisions:
A statistical analysis subsection should be added to the materials and methods section.
line 63 you could add: "Specifically, three cyclin-dependent kinases 4/6, including palbociclib, ribociclib, and abemaciclib, have been approved for the treatment of patients with hormone receptor-positive and human epidermal growth factor receptor 2-negative advanced breast cancer." and cite an article such as: doi: 10.1007/s40264-021-01071-1.
line 79, you could add: "Melanomas are malignant primary tumors originating from melanocytes located in the skin and the mucosal membranes. These malignancies are characterized by high aggressiveness and poor prognosis" and cite an article such as: doi: 10.3390/medicina57040359.
Best Wishes
Author Response
A very interesting original paper demonstrating an essential and isoform-specific role for AKT1 in the induction of senescence by Cyclin-dependent kinase 4/6 inhibition, and in the cGAS-STING function, suggesting the inhibition of AKT1 as a novel therapeutic target. Only minor revisions:
Response: Thank you for your interest in our work.
1. A statistical analysis subsection should be added to the materials and methods section.
Response: We now include a statistical analysis subsection in the methods section (lines 210-215).
2. Line 63 you could add: "Specifically, three cyclin-dependent kinases 4/6, including palbociclib, ribociclib, and abemaciclib, have been approved for the treatment of patients with hormone receptor-positive and human epidermal growth factor receptor 2-negative advanced breast cancer." and cite an article such as: doi: 10.1007/s40264-021-01071-1.
Response: Thank you for the suggestion. We now include in the introduction the mention of these specifically approved CDK4/6i with the suggested citation by the reviewer (lines 67-69).
3. Line 79, you could add: "Melanomas are malignant primary tumors originating from melanocytes located in the skin and the mucosal membranes. These malignancies are characterized by high aggressiveness and poor prognosis" and cite an article such as: doi: 10.3390/medicina57040359.
Response: We appreciate the suggestion, and now include this information about melanoma in the introduction (lines 77-79).